# IFN-Induced PARPs—Sensors of Foreign Nucleic Acids?

**DOI:** 10.3390/pathogens12030457

**Published:** 2023-03-14

**Authors:** Katharina Biaesch, Sarah Knapp, Patricia Korn

**Affiliations:** Institute of Biochemistry and Molecular Biology, Medical Faculty, RWTH Aachen University, Pauwelsstraße 30, 52074 Aachen, Germany

**Keywords:** ADP-ribosylation, MARylation, hydrolase, interferon, macrodomain, PARP, RNA-virus

## Abstract

Cells have developed different strategies to cope with viral infections. Key to initiating a defense response against viruses is the ability to distinguish foreign molecules from their own. One central mechanism is the perception of foreign nucleic acids by host proteins which, in turn, initiate an efficient immune response. Nucleic acid sensing pattern recognition receptors have evolved, each targeting specific features to discriminate viral from host RNA. These are complemented by several RNA-binding proteins that assist in sensing of foreign RNAs. There is increasing evidence that the interferon-inducible ADP-ribosyltransferases (ARTs; PARP9—PARP15) contribute to immune defense and attenuation of viruses. However, their activation, subsequent targets, and precise mechanisms of interference with viruses and their propagation are still largely unknown. Best known for its antiviral activities and its role as RNA sensor is PARP13. In addition, PARP9 has been recently described as sensor for viral RNA. Here we will discuss recent findings suggesting that some PARPs function in antiviral innate immunity. We expand on these findings and integrate this information into a concept that outlines how the different PARPs might function as sensors of foreign RNA. We speculate about possible consequences of RNA binding with regard to the catalytic activities of PARPs, substrate specificity and signaling, which together result in antiviral activities.

## 1. Introduction

In order to establish an innate immune response to invading viruses, cells need to be able to distinguish self from foreign. This is enabled in part by a repertoire of proteins that specifically sense foreign nucleic acids. These proteins belong to the so-called pattern recognition receptors (PRRs) that recognize and bind pathogen-associated molecular pattern (PAMPs), including different pathogen-associated nucleic acids [1,2,3]. In general, upon PAMP binding these PRRs are activated to trigger downstream signaling events via activation of transcription factors, such as interferon regulatory factors 3 and 7 (IRF3, IRF7) and nuclear factor kappa B (NF-κB). This results in the activation of a gene expression program, which includes the induction of interferon (IFN) as well as other cytokine genes. IFNs act in a paracrine and autocrine manner to induce the expression of interferon-stimulated genes (ISGs) by which an antipathogenic state is promoted [1,3].

The nucleic acid-sensing PRRs can be subdivided into two groups, the compartmentalized, endosomal and the cytosolic nucleic acid sensors. A subset of Toll-like receptors (TLRs) belongs to the first subgroup, whereas the second group includes retinoic acid-inducible gene I (RIG-I)-like receptors (RLRs), Protein kinase R (PKR), 2′–5′ oligoadenylate synthetase proteins (OAS1-3), nucleotide-binding oligomerization domain (NOD)-like receptors (NLRs), absent in melanoma 2 (AIM2)-like receptors (ALRs) and cyclic GMP-AMP synthase (cGAS) [2,4,5,6,7,8,9,10].

In addition to these classical PRRs, a growing list of nucleic acid sensor proteins or accessory proteins have been described. These include helicases, ubiquitin ligases and ADP-ribosyltransferases, that can sense certain nucleic acids, assist in, or mediate the recognition of foreign nucleic acids and accelerate downstream signaling thereby contributing to and modulating an antiviral immune response [11,12,13,14,15].

Best known for its viral ribonucleic acid (RNA)-binding activities is PARP13 [11]. In addition, PARP9 has recently been identified as sensor of foreign RNA [15]. For PARP13, RNA binding is facilitated by zinc finger domains, whereas for PARP9 the macrodomain has been identified as viral RNA-binding module. PARP9 and PARP13 belong to the adenosine diphosphate (ADP)-ribosyltransferase diphtheria toxin-like (ARTD) family, of which a small subset of proteins has been linked to innate immunity due to their responsiveness to IFNs (for further reading on *PARPs* as ISGs we refer to a recent excellent review [16]). These proteins share a conserved ADP-ribosyltransferase (ART) domain, which, with exception of PARP13, possesses mono-ADP-ribosylation (MARylation) activity. All these PARPs are characterized with a range of additional protein domains, among them macrodomains, RNA-recognition domains or zinc fingers. Although the functions of these associated domains are largely unknown, many of these have been associated with RNA-binding. Thus, they provide these proteins with the potential ability to function as RNA sensors similar to what has been proposed for PARP9 and PARP13 [11,15]. Together, we hypothesize that IFN-inducible PARPs function as RNA sensing PRRs and expand the RNA-binding modalities of the known classical PRRs with regard to sequence and/or structure specificities. Moreover, also RNA binding might regulate their modes of action and functionality.

## 2. The Classical Pathogen Recognition Receptors

### 2.1. Compartmentalized PRRs

Toll-like receptors involved in sensing pathogenic nucleic acids are TLR3 and TLR7-9 [4,17,18,19]. These TLRs are integrated into the membranes of endosomes with their N-terminal nucleic acid-binding ectodomain facing the inside of these vesicles [4,17,18]. Nucleic acid binding provokes dimerization of two TLRs, upon which diverse signaling processes are initiated [4]. Due to their localization, these TLRs are capable to respond to endocytosed or phagocytosed pathogens that may be disassembled in this compartment through the action of endosomal proteases and nucleases. As a result, pathogen-derived RNA or deoxyribonucleic acid (DNA) are processed and exposed, providing PAMPs that can interact with the endosomal TLRs [18]. This initiates a first wave of antiviral signaling [4,18,19].

To cover the recognition of a broad range of different pathogens, these TLRs have evolved different preferences for nucleic acids [4,18,19]. TLR3 recognizes and binds double-stranded RNA species based on electrostatic interactions between positively charged amino acids as part of the leucine-rich repeats in the ectodomain and the negatively charged ribose-phosphate backbone of the RNA. Binding occurs independently of specific RNA sequences [19]. Recently, its activation by cellular R-loop derived RNA-DNA hybrids has been demonstrated, which provokes subsequent immune signaling resulting in activation of IRF3 has been demonstrated [20]. However, how R-loop processing is regulated and how these hybrids, originally generated in the nucleus, reach the cytosol or even are capable of activating this endosomal receptor remains unclear. Of note is, that R-Loop forming sequence have also been identified among viruses, but whether these indeed form R-Loop structures and are able to trigger TLR3 activation needs to be investigated [21].

TLR7 and TLR8, which are closely related, sense single-stranded RNA and RNA breakdown products. Both, TLR7 and TLR8 harbor two RNA binding motifs, of which the first recognizes a single guanosine or uridine, respectively, whereas the second has been demonstrated to mediate some sequence specificity. TLR7 preferentially binds polyU 3-mers, while TLR8 senses UG/UUG oligoribonucleotides [22,23]. In contrast, TLR9 has been shown to bind to single-stranded CpG motif-containing DNA [4,18].

### 2.2. Cytosolic PRRs

Key sensors of viral nucleic acids in the cytosol, present upon virus infection, are the RLRs [2,7,24]. The eponymous member of these cytosolic receptors is RIG-I. Additional members include melanoma differentiation association gene 5 (MDA5) and laboratory of genetics and physiology 2 (LGP2). All three proteins share a similar domain organization with a central RNA-helicase domain that in concert with their C-terminal domain (CTD) mediates RNA binding [2,7,24]. In contrast to LGP2, RIG-I and MDA5 share two caspase-activation and recruitment domains (CARDs) at their N-terminus that triggers downstream signaling events [2,7]. In case of RIG-I these CARDs are intramolecularly bound by the helicase domain and CTD, provoking a closed conformation of the protein and thereby preventing downstream signaling in absent of a ligand [7,25]. Nucleic acid recognition entails the hydrolysis of ATP by RIG-I and provokes the change to an open conformation and its oligomerization. This allows a closer interaction of the RNA-binding part with nucleic acids while the CARDs are released to interact with mitochondrial interactor of virus signaling (MAVS) for downstream signal transduction [7,24]. This autoinhibitory state shown for RIG-I does not occur for MDA5. Instead, MDA5 rather shows an open and flexible and thus uninhibited conformation. This entails downstream signaling upon overexpression of MDA5 in the absence of an RNA ligand [26,27,28]. Due to the lack of CARDs LGP2 cannot directly initiate downstream signaling via MAVS. But it seems to function as modulator of MDA5 signaling. At low protein levels, LGP2 accelerates and stabilizes MDA5-RNA interaction, whereas high levels of LGP2 lead to MDA5 inhibition [27,29,30].

For all three family members, the recognition of nucleic acids is facilitated by the central helicase domain and the CTD [2,7,24]. These protein domains facilitate the scanning of biochemical features located at the 5’ end of RNA molecules. Despite sharing comparable helicase domains and CTDs, RIG-I and MDA5 sense slightly different features within RNAs [31]. RIG-I prefers shorter double-stranded (ds)RNAs or ssRNAs and is activated by 5’-PPP-dsRNA or 5’-pp-dsRNA, whereas 5’ monophosphorylated RNA stays undetected by RIG-I [32]. Further, RNAs enriched in poly-U/UC or AU regions as well as circular viral RNAs are recognized by RIG-I [33,34,35]. Binding to circular RNAs is proposed to be mediated by RNA structural features or through accessory RNA-binding proteins, which need to be identified [33]. MDA5 preferentially binds to long dsRNAs and AU-rich regions [28,36,37]. LGP2 has been shown to detect a wide range of diverse RNAs. Neither the phosphorylation status of the 5′-end nor the length of the RNA seem to influence recognition and binding by LGP2 [38,39].

RNA sensing by PKR or OAS family proteins 1–3 is also known to contribute to an antiviral defense response [9,10]. PKR recognizes dsRNA molecules longer than 30 bp in a cap-independent fashion [40], but also ssRNA and structured 5’-PPP-RNA binding has been described [41,42]. Binding is facilitated by two tandem RNA-binding domains located in its N-terminal half, which upon RNA binding initiate dimerization of PKR and subsequent kinase activation [43]. OAS1-3 bind to dsRNA [10,44,45,46]. Upon dsRNA binding OAS1-3 synthesizes 2′–5′ phosphodiester-linked oligoadenylates, which serve as second messenger to trigger dimerization and in turn activation of Ribonuclease (RNase) L and thus cleavage of RNA [10,47]. Cleaved RNA fragments serve to amplify antiviral signaling as they are sensed by PRRs [9].

An additional line of immune defense is displayed by NLRs and ALRs [1,6,48]. Upon activation, some NLRs and ALRs have been shown to initiate the assembly of so called inflammasomes, multiprotein enzymatic complexes in which they oligomerize and bind to apoptosis-associated speck-like protein containing CARD (ASC) domains to mediate the proteolytic activation of caspase-1. This in turn enables the maturation of cytokines such as Interleukin 1β (IL-1β) and IL-18, thereby contributing to an antiviral immune response.

Among the NLRs, NLRP3 has been shown to be activated by a broad range of diverse RNAs [8,49]. However, direct interaction with RNAs has not been demonstrated. Instead, NLRP3 assembles with accessory proteins, among them DExD/H-box RNA helicases or TRIM ubiquitin ligases, which have been shown to enable RNA-sensing and subsequently the activation of the inflammasome [8,49]. In contrast to NLRP3, AIM2 as representative of the ALRs is activated by DNA [6,48,50].

In addition to AIM2, cGAS functions as cytosolic sensor of DNA [5]. Full activation of cGAS occurs upon binding to longer DNA molecules. These allow for dimerization of cGAS, a prerequisite for full activation. However, cGAS has been shown to recognize a variety of DNA molecules, among them dsDNA, ssDNA providing secondary structures that result in dsDNA, or RNA-DNA hybrids (as e.g., derived from R-loops). Upon binding, signaling is propagated through cGAMP-mediated activation of stimulator of interferon genes (STING), resulting in the activation of IRF3 [5]. Thus, cGAS can be activated by pathogenic DNA but also by cellular DNA, for example in response to cytosolic DNA as a result of missegregation of chromosomes, micronuclei and DNA shattering [51].

Besides these classical PRRs, several additional host factors have been identified serving as sensors for foreign nucleic acids, among them DExD/H box helicases, trispartite motif family (TRIM) ubiquitin ligases and a growing number of various additional RNA-binding proteins [12,13,14,52,53]. Also, the very heterogeneous family of scavenger receptors has been implicated in innate immunity and some members have been shown to bind to foreign nucleic acids [54]. For some of these RNA-binding proteins scaffolding function has been implicated [3,5,12]. They sense and bind foreign RNA, and present it to RLRs, thereby contributing to and amplifying antiviral signaling [3,5,12].

## 3. PARP13—A Sensor of Viral RNA

One of these scaffolding proteins referred to above, which is involved in the innate immune response, is the zinc finger antiviral protein (ZAP), also known as PARP13 (Figure 1). Even though it does not possess catalytic activity, it is known for its efficient antiviral activity [11]. PARP13 exists in four different isoforms, arising from alternative splicing and polyadenylation [11,16]. The two best studied isoforms are PARP13.1 (ZAPL) and PARP13.2 (ZAPS), the latter lacks the PARP-like domain [11]. While PARP13.1 seems to be constitutively expressed, PARP13.2 is induced upon interferon signaling [55]. An interaction with the 3′ untranslated region (3′UTR) of the interferon messenger RNA (mRNA) has been described for PARP13.2, which is therefore considered to be involved in a negative feedback response to IFN signaling [55]. Interestingly, PARP13.2 was found to colocalize with RIG-I when stimulated with 5′-PPP-double stranded RNA (dsRNA) and appears to play a role in promoting interferon production [56].

All isoforms of PARP13 have an RNA-binding domain (RBD) consisting of four CCCH zinc finger (ZnF) motifs, the second of which is known for its hydrophobic binding pocket with a high affinity for CpG-dinucleotides [11,57]. The other ZnFs display weak affinity for RNA of unknown sequence [11]. PARP13 is able to dimerize, and even multimerization of PARP13 on target RNA has been suggested for efficient defense against pathogens [11].

Recently, a severe acute respiratory syndrome coronavirus type 2 (SARS-CoV-2) RNA interactome screen identified PARP13, as well as its cofactor TRIM25, to bind directly to the viral RNA [58]. Following ectopic expression of PARP13.1 and PARP13.2, PARP13.2 but not PARP13.1 appeared to have an antiviral effect, as evidenced by a significant reduction in SARS-CoV-2 non-structural protein 12 (nsP12) RNA levels, encoding the viral RNA-dependent RNA polymerase [58]. In contrast, Nchioua and colleagues reported a reduction in the accumulation of SARS-CoV-2 full length RNA only in PARP13.1 overexpression experiments [59]. However, for both isoforms a reduction in the RNA levels of SARS-CoV-2 structural spike- and nucleocapsid protein was observed [59]. Differences in cellular localization might account for these findings, as PARP13.2 has a diffuse cytoplasmic distribution, while PARP13.1 can be S-farnesylated, which localizes it to endolysosomes or the endoplasmic reticulum (ER) [11]. SARS-CoV-2 forms ER-derived double-membrane vesicles for replication [60]. Indeed, it was later demonstrated that S-farnesylation of PARP13.1 is needed for SARS-CoV-2 attenuation [61]. Antiviral activity has also been described against influenza A virus (IAV). While PARP13.1 seems to modulate viral protein expression, PARP13.2 has been described to directly target IAV RNA [11,62]. Liu and colleagues reported PARP13.1 to promote the poly-ADP-ribosylation (PARylation) of IAV polymerase proteins, which leads to their subsequent ubiquitination and degradation [62]. However, as PARP13 has no reported catalytic activity, another ADP-ribosylating protein needs to be involved in this process. The shorter isoform, PARP13.2, is able to bind to IAV basic RNA polymerase 2 (PB2) mRNA and leads to its degradation as well as preventing its translation [63]. This process is counteracted by the non-structural protein 1 (NS1) protein of IAV, which was found to prevent viral RNA binding by PARP13.2 [63]. Interestingly, also the NS1 mRNA seems to be unaffected by PARP13.2 [63]. Potentially, this could be attributed to secondary structures within the NS1 RNA, which has been demonstrated to form hairpins resulting in large parts of this RNA being double stranded [64]. Another genus of viruses restricted by PARP13 are alphaviruses like Sindbis virus, which is targeted by PARP13.1 in stress granules (SGs) [55].

Recently, different groups found in crystallization experiments that the WWE2 pocket of PARP13 is able to bind to an ADP-ribose (ADPr)-moiety of poly-ADP-ribose (PAR) chains [65,66]. Xue and colleagues also confirmed these results in vitro and revealed an essential role of two amino acids in the WWE2 domain, W611 and Q668, for this binding. Further, they demonstrated that the ZnF5, WWE1 and WWE2 of PARP13 combine to form a domain they termed central domain (CD), and that this CD binds to PAR in cells. The long isoform of PARP13, PARP13.1, was also shown to bind PAR in cells, although not as efficiently as the isolated CD [66]. This binding plays an important role in stress granules (SGs), where PAR binding is a prerequisite for PARP13-CD and PARP13.1 re-localization [66]. In addition, mutational impairment of PARP13.1 binding to PAR was found to attenuate its antiviral activity [66]. Localization to stress granules has also been described for PARP13.2, which is increasingly PARylated upon stress [67]. Thus, stress granules (SGs) allow the accumulation of RNA, PAR and PARP13 [66,68]. Whether clustering contributes to the antiviral activity of PARP13, namely promoting RNA degradation or inhibiting translation will need to be addressed. Worth mentioning is, that similar to PARP13 additional PARP proteins have been shown to associate to SGs, suggesting a concerted action or a synergistic role of PARPs in SG formation and/or function. Pointing to a similar direction is the finding, that PARP13, although lacking catalytic activity itself, is ADP-ribosylated and therefore must closely interact with other PARP enzymes [67]. This ADP-ribosylation may control PARP13 function as e.g., shown for PARP7, which MARylates cysteine residues in ZnFs, thereby interfering with the ability of PARP13 to interact with RNA [16]. We expect may additional interaction between PARP proteins as well as other PRRs and downstream effectors. Thus, how PARPs synergize for efficient recognition of nucleic acids and defense against pathogens are exciting questions in the field of innate immune defense.

## 4. The IFN-Regulated Subclass of PARPs

### 4.1. The PARP Family

Based on domain organization and structural analysis PARP13 is assigned to the family of ADP-ribosyltransferases diphtheria toxin-like (ARTDs), which in total encompasses 17 members [69,70,71]. They all share a highly conserved ART domain, which with exception of PARP13 enables these proteins to catalyze ADP-ribosylation. ADP-ribosylation is a reversible posttranslational modification (PTM), which is characterized by the addition of one or several ADP-ribose moieties onto a substrate [70]. Based in part on the amino acid composition of the catalytical triade individual enzymes can either catalyze PARylation (PARP1, PARP2, TNKS1 and TNKS2) or MARylation (PARP3, PARP4, PARP7-PARP12, PARP14-PARP16) [70,72]. To do so, they consume nicotinamide adenine dinucleotide (NAD^+^) as a cofactor and transfer ADP-ribose, either a single moiety (MARylation) or in an iterative process (PARylation) multiple units with release of nicotinamide [70]. PARP13 is the only family member lacking ADP-ribosylation activity due to its inability to properly bind NAD^+^ [73].

In the following we will concentrate on the interferon-responsive PARPs (PARP9-15; Figure 1) [16], MARylation and the (potential) nucleic acid sensing capabilities of this subset of PARPs.

### 4.2. Regulation and Propagation of MARylation

Like other PTMs MARylation needs to be read and the signal propagated. The macrodomains 2 and 3 of PARP14 have been identified as readers of MARylation [70,74,75]. Further, MARylation displays a fully reversible PTM enabled by the hydrolytic activity some macrodomains possess [70]. Cellular erasers of MARylation include MacroD1, MacroD2 and TARG1. De-MARylation is enabled by their active macrodomain [70]. The macrodomain fold is highly conserved among all species of life and is embedded in non-structural proteins of several positive sense single-stranded ((+)ss) RNA viruses as well [16,76,77].

The induction of MARylating PARPs by the innate IFN system in combination with the ability of several viral macrodomains to revert MARylation indicates an antiviral role of IFN-inducible PARPs. Further, it has been shown that PARPs have evolved under strong positive selection, additionally pointing to a function in innate immunity [78,79].

However, insights into mechanisms and the exact function of IFN-inducible PARPs remain elusive. One possibility how IFN-inducible PARPs might contribute to an antiviral response is by recognition of foreign nucleic acids. As outlined before, adaptor proteins like DExD/H box helicases or PARP13 can serve as scaffolds bringing nucleic acids and effector proteins in close proximity. Similarly, the IFN-responsive PARPs could function as scaffolds thereby assisting in RNA-recognition by one of the classical PRRs. On top of that, their MARylation activity could add another level of regulation to fine tune the innate immune response. There are indications that the presence of viral RNA might trigger MARylation activity of these enzymes [80,81]. Postulating that RNA binding determines catalytic activity, it might also allow for redirecting catalytic activity to distinct substrates. These could be both viral and host factors. Moreover, the altered specificity might also affect protein stability, for example by reducing automodification, thereby conferring stability of certain PARP enzymes [82]. Additionally, viral RNA might represent a substrate for MARylation, as RNA has been identified to be MARylated both in vitro and in cells [83,84].

### 4.3. Domain Organization of IFN-Regulated PARPs

Of note is, that the IFN-responsive PARPs all display domain and motifs potentially implicated in nucleic acid binding (Figure 1).

**Figure 1 pathogens-12-00457-f001:**
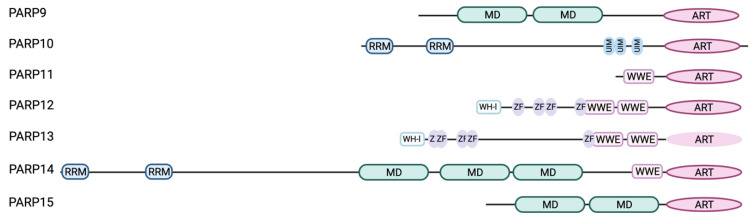
Domain architecture of the IFN-responsive PARPs. All IFN-responsive PARP family members contain the conserved ADP-ribosyltransferase (ART) domain at their C-terminus. Except for PARP13 the ART domain of the other PARPs possesses MARylation activity [72,73]. PARP9, PARP14 and PARP15 contain macrodomain (MD) repeats, either 2 as in case of PARP9 and PARP15 or three as seen for PARP14. In addition to the three macrodomains PARP14 is also equipped with two RNA-recognition motifs (RRM) at its N-terminus, known to mediate RNA-binding. Similarly, PARP10 carries two RRMs at its N-terminus. PARP11-PARP14 harbor one (PARP11, PARP14) or two WWE (PARP12, PARP13) modules, known to facilitate poly-ADP-ribose binding. N-terminally PARP12 and PARP13 both contain Winged-Helix-like (WH-l) DNA-binding domains followed by five zinc finger motifs (ZF), known to mediate binding to nucleic acids. PARP10 is unique, as it is the only family member equipped with ubiquitin-interaction motifs (UIMs), of which it carries three in its C-terminal half (Created with BioRender.com).

PARP12 resembles the overall domain structure of PARP13 and similarly is equipped with several ZnFs. These domains are well described as nucleic acid-binding modules, among other functions, and as such broadly involved in host-pathogen interactions [85]. This provokes questions as to which function(s) can be assigned to the ZnFs of PARP12 and whether these are implicated in RNA sensing.

There is accumulating evidence that macrodomains represent an additional nucleic acid-binding module. Recently, PARP9 has been shown to bind to viral RNA mediated by its first macrodomain [15]. The capability of binding to RNA has been demonstrated for TARG1 as well [86]. The macrodomain as binding module for nucleic acids has also been established from findings with some viral macrodomains (vMDs). The vMD of Chikungunya virus (CHIKV) or Venezuelan encephalitis virus (VEEV) have been shown to bind ssRNA [87], whereas the second and third vMD (SARS unique domains, SUD) of SARS-Coronavirus have been demonstrated to bind G-quadruplexes [88,89]. Besides PARP9, PARP14 and PARP15 belong to the macrodomain-containing IFN-stimulated PARPs. Whereas PARP14 macro2 and macro3 as well as PARP15 macro2 have been identified to bind to MAR [75,90], the function of the first macrodomain within both proteins remains elusive. However, based on sequence comparisons they are phylogenetically closer related to the hydrolytic macrodomains encoded by ssRNA viruses, maybe allowing to postulate an ability in RNA binding as well (Figure 2).

In addition to its macrodomains, PARP14 displays two RNA-recognition motifs (RRMs) near its N-terminus, which are separated by an intrinsically disordered region (IDR, according to the amino acid sequence analysis using PONDR) from its other functional domains. This is also the case for PARP10 (analysis by PONDR) (Figure 1). RRMs but also IDRs individually or cooperatively can mediate RNA binding [91,92,93]. Generally, multiple RRMs work in tandem thereby facilitating proper RNA binding and conferring RNA specificity [94]. It will be of interest to evaluate nucleic acid binding modes of these subset of PARPs. Do these domains indeed sense foreign nucleic acids to contribute to a robust antiviral response?

**Figure 2 pathogens-12-00457-f002:**
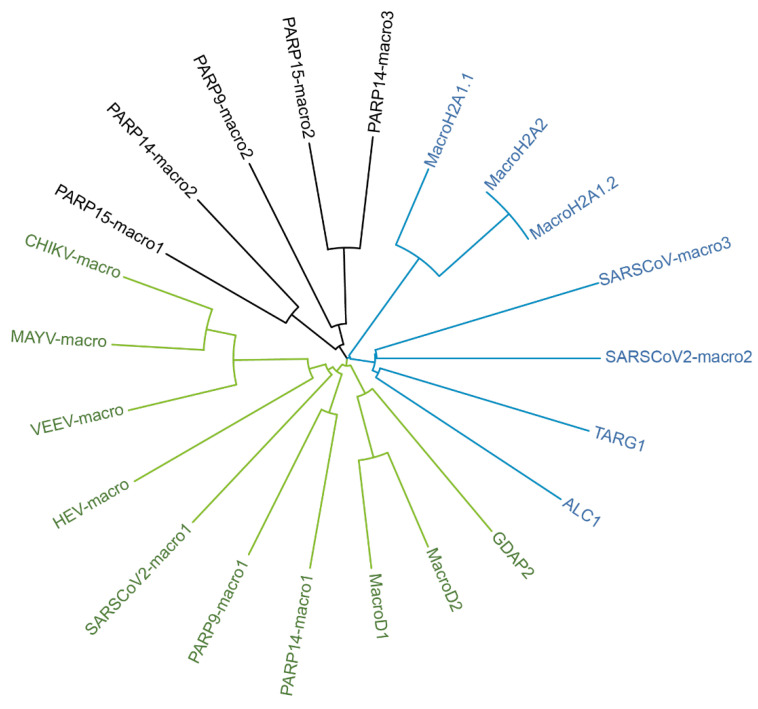
Phylogenetic tree of human and some selected viral macrodomains. Amino acid sequences (>sp|O75367|184-370_MacroH2A1.1; >sp|Q9P0M6|184-370_MacroH2A1.2; >sp|Q9P0M6|184-370_MacroH2A2; >sp|Q86WJ1|704-897_ALC1; >sp|Q9Y530|2-152_TARG; >sp|Q8IXQ6|107-296_PARP9-macro1; >sp|Q8IXQ6|306-487_PARP9-macro2; >sp|Q460N5|791-978_PARP14-macro1; >sp|Q460N5|1003-1190_PARP14-macro2; >sp|Q460N5|1216-1387_PARP14-macro3; >sp|Q460N3|78-267_PARP15-macro1; >sp|Q460N3|293-464_PARP15-macro2; >sp|Q9BQ69|141-322_MacroD1; >sp|A1Z1Q3|59-240_MacroD2; >sp|Q9NXN4|43-223_GDAP2; >sp|P36328|1330-1489_VEEV-macro; >sp|Q8JUX6|1334-1493_CHIKV-macro; >sp|Q8QZ73|1335-1493_MAYV-macro; >sp|P0DTD1|1025-1194_SARSCoV2-macro1; >sp|P0DTD1|1231-1359_SARSCoV2-macro2; >sp|P0DTD1|1367-1494_SARSCoV-macro3; >sp|Q9WC28|775-921_HEV-macro; >sp|K9N7C7|1110-1276_MERS-macro1; >sp|K9N7C7|1278-1404_MERS-macro2) were analyzed by CLUSTAL 2.1 and the phylogenetic tree file uploaded to iTOL 6.6 to generate this phylogenetic tree [95].

### 4.4. IFN-Regulated PARPs as Host Restriction Factors

As already stated, PARP12 possesses a similar domain organization as PARP13 but its ART domain displays enzymatic activity [16] (Figure 1). While PARP13 is already known for its role as a PRR in the innate immune response, a similar function might be postulated for PARP12 [11,96]. However, for PARP12 RNA binding has not been confirmed so far experimentally, but there is evidence coming from PARP12 being recruited to SGs [67,97,98]. SGs are condensates enriched in mRNA due to the stress-dependent stalled translation complexes and PAR [67,99]. Localization of PARP12 to these condensates is dependent on its ZnFs and WWE domains, suggesting that the ability to potentially bind both RNA and PAR provokes PARP12 to localize to SGs [97,98]. Of note is, that like RNA binding, PAR-binding by the WWE domain of PARP12 has not been experimentally validated. A functional role of PARP12 in SG biology granules has yet to be found, but as SGs are discussed as first line response to viral infections, the regulation and/or modulation of these condensates might be one mode of antiviral action of PARP12 [100]. It is worth pointing out is that in addition to PARP13 and PARP12, PARP14 and PARP15 have been identified as SG proteins as well, at least when overexpressed [67]. It will be interesting to analyze whether PARP12, analogous to PARP13, regulates RNA turnover and/or translation and whether this is restricted to viral RNAs or might also be relevant for host mRNAs in infected and thus stressed cells. An additional line of evidence for PARP12 as RNA-binding protein is deduced from recent SARS-CoV-2 research. The identification of host factors interacting with the SARS-CoV-2 RNA genome revealed PARP12 and PARP13 as interacting proteins [58,101].

Indeed, PARP12 has been identified as restriction factor for some viruses [81,102,103]. One potential mechanism being discussed is limiting alphavirus replication by modulation of cellular translation [102]. Upon VEEV infection, PARP12 seems to complex with ribosomes and several proteins known to play a role in translation [102]. This might also provide a link to SG biology and/or the modulation of these condensates as they are enriched in stalled translation complexes [100]. In addition, PARP12 limits Zika virus (ZIKV) replication in fact upon interaction with PARP11 via their WWE domains [104,105]. Here, the restrictive effect is mediated by promoting PARylation of the viral non-structural proteins NS1 and NS3 targeting them for proteasomal degradation [104,105]. This resembles the mode of action shown for PARP13.1 with regard to IAV proteins being primed by PAR for proteasomal degradation [62]. Again, presumably other PARP enzymes are also involved in this process, as PARylation is neither catalyzed by PARP12 nor PARP11 [72].

PARP11 has been identified as regulator of IFN signaling. It has been shown to catalyze MARylation of β-TrcP, a ubiquitin E3 ligase. This results in the subsequent ubiquitination and turnover of the IFNα/β receptor 1 (IFNAR1) indicating a feedback control of IFN signaling by PARP11 [106].

PARP9, along with PARP14 and PARP15, is one of the macrodomain-containing PARPs [16] (Figure 1). However, to date it has not been fully elucidated for PARP9, whether or not it has ADP-ribosylating activity [16]. The PARP9 macrodomains have been identified to bind PAR enabling PARP9 colocalization with the PARylating enzyme PARP1 upon DNA damage [107,108]. Furthermore, an antiviral role for PARP9 has been discussed. In dendritic cells, influenza A, a minus-strand RNA virus, induces the expression of PARP9 [15]. Further, Xing and colleagues reported a protective effect of PARP9 against minus-sense RNA virus vesicular stomatitis virus and dsRNA reovirus infection in mice, whereas this effect does not occur with the DNA-virus Herpes simplex virus type 1 (HSV-1) [15]. They found the first macrodomain of PARP9 to be essential for binding of viral dsRNA ranging from 1100 base pairs (bp) to 1400 bp (Table 1). Furthermore, PARP9 contributes substantially to the type-I IFN production by activating the phosphoinositide-3-kinase/protein kinase B (PI3K/AKT) signaling pathway [15].

For many processes, however, PARP9 forms a heterodimer with the E3 ubiquitin ligase deltex E3 ubiquitin ligase L (DTX3L). Together they play a role in DNA damage repair and antiviral defense [15,108]. The DTX3L/PARP9 heterodimer is capable of selectively MARylating ubiquitin [108]. The authors suggest that this modification depends on the catalytic activity of PARP9 [108]. Russo and colleagues found that the DTX3L/PARP9 heterodimer plays a central role in ADP-ribosylation induced upon induction of ISGs. This seems to be independent of PARP9 activity itself, suggesting a potential crosstalk with other MARylating PARPs or a concerted action of these proteins. The increase in overall MARylation is reversed by the hydrolase activity of the SARS-CoV-2 nsP3 macrodomain1 [109,110].

In 2016, Iwata and colleagues found signal transducer and activator of transcription 1 (STAT1) and STAT6 to be ADP-ribosylated in vitro by PARP14, a process suppressed by PARP9. They further claimed STAT1α phosphorylation to be inhibited by PARP14 mediated STAT1α ADP-ribosylation [111]. Additionally, an anti-inflammatory role of PARP14 in macrophages, promoting the interleukin (IL)-4 response and suppressing IFN-γ induced responses, was observed [111]. Although this work has received strong criticism [112], at least the PARP9-PARP14 interaction has been confirmed in co-immunoprecipitation experiments by other groups [113]. Grunewald and colleagues suggest that PARP14 can regulate the IFN response both, dependent on ADP-ribosylation, but also independent of its catalytic activity [114]. Further, they observed increased viral replication of mouse hepatitis virus (MHV) in Parp14 inhibition and knockdown experiments, suggesting antiviral capacities of PARP14 [114]. In viral crosslinking and solid-phase purification (VIR-CLASP) experiments for Chikungunya virus (CHIKV), PARP14 and PARP9 were identified as CHIKV-RNA interactors [115]. A screen for interactors of the IAV-genome in contrast, did not reveal interaction of any of the mono-ARTDs [115].

PARP14 has three macrodomains and macro2 and macro3 have been reported to bind to MARylated PARP10 but seem to lack hydrolase activity and therefore are considered as readers of MARylation [75]. Interestingly, the PARP14 macrodomain1 has been described to resemble, at least at the sequence level, the SARS-CoV-2 macrodomain (Figure 2) [116,117]. PARP14 is the largest of the PARP enzymes and has an RNA recognition motif (RRM) at its N-terminus followed by a long intrinsically disordered region, the function of which are as yet unknown [118].

PARP14 binds to the 3’UTR of tissue factor mRNA in synergy with tristetraprolin (TTP) upon Lipopolysaccharide (LPS) stimulation (Table 1) [119]. However, which domains of PARP14 are involved in this interaction or if PARP14 mediated ADP-ribosylation contributes to this interaction remains to be determined [119]. Nucleic acid binding of PARP14 has also been reported by Riley and colleagues, who found two putative DNA motifs recognized by PARP14 (Table 1). These motifs are present in the promoter region of *interleukin-4 (Il-4)* and *Il-5* and PARP14 seems to have a role in the expression of T helper type 2 (Th2) cytokines [120]. This is further supported by findings of a role of PARP14 in allergic reactions in mice [121].

**Table 1 pathogens-12-00457-t001:** Overview on RNA-binding modalities of the classical PRRs and the IFN-regulated PARPs.

Protein	RNA	Reference
**TLR3**	double-stranded RNA; sequence independent	[4,17,18,19,20]
**TLR7**	single-stranded RNA and RNA breakdown products; preferentially binds polyU 3-mers	[4,17,18,19,23]
**TLR8**	single-stranded RNA and RNA breakdown products; recognizes UG/UUG oligoribonucleotides	[4,17,18,19,22]
**RIG-I**	5′-PPP-dsRNA or 5’-pp-dsRNA;RNAs enriched in poly-U/UC or AU regions; circular viral RNAs	[2,7,24,31,32,33,34,35]
**MDA5**	long dsRNAs; AU-rich regions	[2,7,24,28,31,36,37]
**LGP2**	range of diverse RNAs	[38,39]
**PKR**	dsRNA > 30 bp; ssRNA; 5′-PPP-RNA	[9,40,41,42]
**OAS1-3**	dsRNA	[9,10,44,45,46]
**DExD/H box helicases**	Adapter proteins; enables RNA sensing and activating of PRRs	[13,53]
**TRIM ubiquitin ligases**	Adapter protein; enables RNA sensing and activating of PRRs; preferentially binds to positive strand RNAs	[14]
**PARP13**	ssRNA (CpG bound by ZnF2); weak binding of RNA (of unknown sequence) by ZnF 1+3+4	[11,57]
**PARP9**	Macrodomain: viral dsRNA binding ranging from 1100 base pairs (bp) to 1400 bp	[15]
*PARP10*	*RRMs potentially mediate RNA-binding*	
*PARP11*	Unknown	
*PARP12*	*ZnFs potentially mediate interaction with host and viral RNA*	
**PARP14**	Binds some host mRNAs via 3′UTR;two putative DNA-motifs bound by PARP14 (Motif 1: CACTGAGTGGAG; Motif 2: TCCAAGGATC)*RRMs and macrodomains potentially mediate interaction with host and viral RNA*	[119,120]
*PARP15*	*Macrodomains potentially facilitate RNA binding*	

PARP14 was found to be localized mainly in the cytosol and translocates to the nucleus upon LPS treatment [113]. It also seems to be involved in the translocation of other proteins to the nucleus, especially those that are IFN inducible [113].

PARP10 is highly expressed in hematopoietic cells, supporting a functional role in innate immunity [122]. Like PARP12, PARP10 has been shown to be restrictive for viral replication [81,102,103]. Atasheva and colleagues showed that expression of PARP10 from a second subgenomic promotor within the VEEV genome results in translation inhibition [102]. However, how PARP10 interferes with translation remains open. Similarly, whether this possible modulation of translation confers to its antiviral activity is unclear.

Recently, non-structural protein (nsP) 2 of CHIKV has been identified as PARP10 substrate. MARylation impairs proteolytic activity of nsP2, which is essential for replication [81]. CHIKV nsPs are translated as polyprotein in need to be processed into the individual nsPs, which subsequently form the functional replication complex [123]. Thus, the antiviral activity of PARP10 might be mediated at least in part by modification and regulation of viral proteins.

Interestingly, MARylation of CHIKV-nsP2 was only observed when mimicking a viral infection by transfection of an in vitro transcribed RNA replicon. Plasmid-based co-expression of GFP-nsP2 and PARP10 was not sufficient to induce MARylation [81]. Similar results were observed studying ADP-ribosylation in context of an infection with the murine hepatitis virus (MHV), a coronavirus. The nucleocapsid (N) protein of MHV was only found to be ADP-ribosylated upon MHV infection and failed modification when expressed exogenously in cells [80]. These findings foster speculation. Is the presence of viral RNA necessary for full activation of PARP10 as well as other PARPs?

N-terminally PARP10 possesses two RRMs near the N-terminus. This is followed by an intrinsically disordered, glycine-rich domain (Figure 1). Whether these enable nucleic acid binding needs to be investigated to address the question whether PARP10 might function as PRR.

As pointed out above, RNA has been identified as substrate for MARylation [83,84,124,125]. The isolated catalytic domains of PARP10 as well as PARP15 are capable to MARylate the terminal 5′ phosphate of ssRNA in vitro. However, the full-length variants of these proteins failed to do so in vitro [83,84]. The ADP-ribosyltransferase identified to MARylate RNA as full-length protein in vitro and in cells, is TRPT-1. MARylation of 5′-P-RNA has been shown to prevent translation [84].

### 4.5. Perspective on IFN-Regulated PARPs as Sensors of Viral RNA

What can be drawn from these findings? Quite clearly, PARPs are involved in antiviral defense. There is increasing evidence linking this subset of IFN-responsive PARP enzymes to innate immunity, as summarized in recent reviews [16,109,118]. But as this is quite an emerging and rapidly developing research field, there are ample open questions to be addressed and answered.

Besides induction by IFNs, we hardly understand how the expression of these PARP genes and the function of the encoded proteins are regulated. How is their catalytic activity regulated? Is a precise regulation of MAR activity needed? How is turnover of these proteins achieved? What are the functions of the diverse protein domains these PARP proteins are equipped with? Is there crosstalk between these different domains and, extending on this, do they provide functionality separated from MAR activity? Further, how do the individual enzymes synergize to contribute to the establishment of a robust antiviral response? What are substrate molecules (protein or nucleic acids) to fine tune an immune response to one or the other pathogen? How is specificity achieved? In this last section we like to speculate on possible answers to these questions.

Based on their domain organization (Figure 1), we speculate that this subset of PARPs interacts with foreign but possibly also cellular nucleic acids. Viral RNAs exhibit a lot of secondary structures, which along with sequence and/or modification of the RNA might allow for recognition and binding [126,127]. These complex secondary structures, mostly located in the 5’UTR and 3’UTR of viral RNAs shield them from recognition by many ssRNA sensors [128]. In addition, viruses have evolved different strategies, such as cap-snatching (stealing the cap from host mRNA) or cap-imitation to evade recognition by the classical PRRs [129]. Thus, it is conceivable that PARPs, such as PARP9, come into play (Figure 3). By binding RNA, they might assist in activation of the classical PRRs, as has been shown for PARP13 in concert with RIG-I [11].

A direct link to inflammasome activation has not been elucidated yet, but NLRP3, which is activated upon a broad range of RNAs does fully rely on accessory proteins, as it has no intrinsic RNA-binding capability [49]. In such scenarios PARPs could come into play to sense nucleic acids and as a consequence bind to and mediate PRR activation. This might be controlled by MARylation. Indeed, ADP-ribosylation of NLRP3 has already been shown. PARylation by PARP1 contributes to its activation and subsequent inflammasome assembly [130]. Further MARylation of NLRP3 by bacterial toxins has been demonstrated to contribute to inflammasome activation [131]. It will be interesting to test whether IFN-regulated PARPs might bridge RNA-sensing and inflammasome activation and whether this is independent on MARylation.

RNA-binding might trigger the activation of PARP enzymes and contribute to specificity, suggested by findings with CHIKV-nsP2 or the N-protein of MHV (Figure 3) [80,81]. In these studies, modification of the viral proteins could only be observed after infection and thus presence of viral RNA. The concept of nucleic acid dependent enzyme activation has long been known for PARP1, which is fully activated only upon presence of nicked DNA due to the crosstalk between ZnF III and the ART domain [132]. Such domain crosstalk is well imaginable for the IFN-regulated PARPs as well. Another mode of activation, although highly speculative at present, might be comparable to how RIG-I is activated [25]. The RRMs and the long intrinsically disordered glycine-rich region present in PARP10 and PARP14 might contribute to an inactive conformation, which opens when the proteins interact with RNA (Figure 3). Such a more open conformation might then allow catalytic activity and/or recognition of substrates. Thus, it will be interesting to clarify whether such intramolecular interactions occur and how these are regulated.

Further, promiscuity of PARP enzymes has been discussed recently [133]. Promiscuity might be overcome by co-factors. For example, HPF-1 directs PARP1 activity towards modification of serine and DTX3L has been discussed to confer to PARP9 catalytic activity [108,134]. An interesting idea is, that in addition to proteins acting as co-factors, RNA might also convey specificity thereby shifting a potential repertoire of substrates (Figure 3). Thinking this further, RNA binding might also result in specific substrate modification instead of automodification. Moreover, inhibition of catalytic activity of some PARPs was shown to increase their stability, indicating that automodification provokes their proteasomal degradation [82]. Thus, RNA-binding of these PARPs might reduce automodification due to the changes in substrate specificity, thereby promoting stability of IFN-responsive PARPs. This increase in protein might be important to enhance the cellular capacity to recognize pathogen nucleic acids. Moreover, once the infection stress is resolved and foreign RNA is eliminated from the cells, PARPs would switch back to automodification, promoting their degradation. Thus, such a scenario would initially enhance and subsequently participate in the timely turn-off of the innate immune response, thus preventing toxic effects due to the fact of overshooting immunity.

The RNA-binding capacities of PARP-enzymes might interfere with viral translation. Alphaviruses, for example, contain high CpG-content and are therefore recognized and targeted by PARP13 [129]. PARP13 in turn was shown to interact with eukaryotic translation initiation factor 4G (eIF-4G) and eIF-4A [129]. Macrodomain-associated PARPs might interfere with SARS-CoV-2 RNA translation. The SARS-CoV-2 nsP3 localizes to ER-derived double membrane vesicles [60]. The SUD of nsP3, consisting of two viral macrodomains and the domain preceding ubiquitin-like 2 (Ubl2) and papain-like protease 2 (PL2^pro^) (DPUP), has been shown to interact with ribosomes and polyadenylate-binding protein-interacting protein 1 (PAIP1) [128]. This interaction is thought to be crucial for viral translation. Furthermore, the macrodomains in the SUD are known to be capable of binding G-quadruplexes and, in the case of macro3, probably poly-A [128]. The binding of viral RNA to the nsP3 SUD macrodomains, could also shield them from recognition by human macrodomains. However, this assumption is still rather vague, as it has not yet been shown that viral RNA binds to the CoV-2 SUD MDs, nor that the human MDs would be able to engage with the viral RNA here. Alternatively, the viral macrodomains might bind to host mRNAs and thus hinder their translation together with nsP1 [128]. However, in both cases it is also interesting to note the proximity of the viral macro1 adjacent N-terminal to the SUD. Macro1 has hydrolase activity, suggesting that PARPs or ADP-ribosylation are involved in attenuating viruses by interfering with their translation [135].

Viral RNA itself could be a substrate (Figure 3). In vitro studies failed to show MARylation by full length PARP10 or PARP15 [84]. However, given the artificial nature of the 5’-P-RNA-stretch used in these in vitro experiments, modification of RNA by PARP enzymes cannot be excluded. Again, structural and/or sequence motifs of RNA might be important for binding and for altering activity and/or specificity of MARylation, aspects to be elucidated by future research.

Interaction and collaboration between PARPs could also be mediated by RNA. Several of these PARPs, at least when overexpressed appear to form condensates in cells [136]. RNA plays an important role as scaffold in many condensates. MARylation might in addition to RNA allow for recruiting these PARPs to such condensates. Based on studies with TARG1, RNA binding and APD-ribose binding appear to be not exclusive, suggesting that macrodomains might well be capable of recognizing MAR signals as well as RNA at the same time [86].

After this rather speculative ideas on PARPs as sensors of foreign RNA and potential consequence of this RNA-interaction, there is still one obvious question to be answered. Do the IFN-regulated PARPs need tight regulation? As they are involved in innate immunity, does deregulation or activating mutations link PARPs to autoimmune disorders? Of note is also that PARP enzymes might act as double-edged sword, not only playing an antiviral role but also being exploited by some viruses. One such candidate might be PARP11, as counterbalance for IFN signaling [106].

## 5. Conclusions

The last years have created several lines of evidence indicating a subset of the MARylating ARTDs plays a role in innate immunity. With proteins and recently also RNA being substrates of MARylation potential mechanisms how they confer to a robust antiviral response are being discussed and evaluated. In addition to the ART domain and modulation by catalytic activity, potential roles of various other domains, with which the IFN-regulated PARPs are equipped, are coming in focus for their possible contributions to antiviral activities. Certainly, we are far away from understanding the necessary details of the functions of these PARP proteins to draw a comprehensive picture of their involvement in innate immunity. Nevertheless, in this review we present possibilities how the additional domains besides the ART domain might contribute to innate immune signaling. Obtaining a more complete understanding of their functions and interplay with viral and host factors, both protein and RNA, will most certainly define novel starting points for pharmacological intervention.

## Figures and Tables

**Figure 3 pathogens-12-00457-f003:**
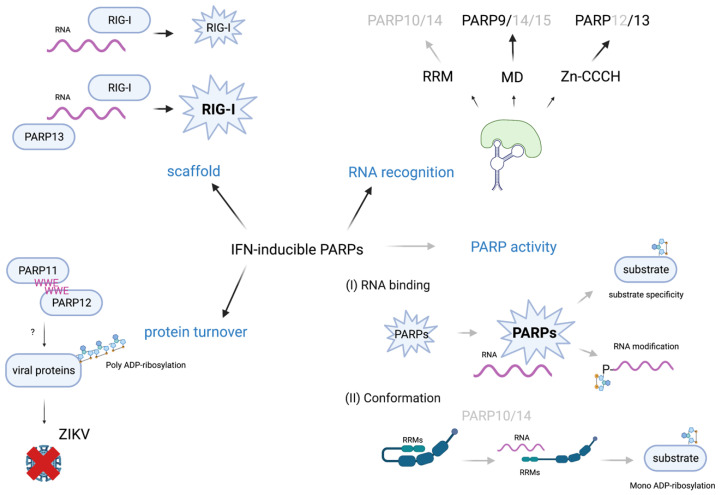
IFN-regulated PARPs as sensors of foreign RNA and possible consequences of this interaction. IFN-inducible PARPs might play a role in foreign nucleic acid recognition through specific protein domains, such as RNA-recognition motifs (RRMs), macrodomains, or CCCH zinc fingers. Similar to PARP13, these PARPs might act as scaffolds, presenting RNA to classical PRRs thereby accelerating downstream signaling. RNA binding might trigger MARylation activity, maybe by a conformational switch, allowing for specific substrate modification. A concerted action of PARPs and interaction between them might contribute to their antiviral activities, as already shown for PARP11 and PARP12, which contribute to degradation of viral proteins. (Black: known mechanisms; grey: speculative but possible mechanisms; created with BioRender.com).

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
