# Peer review of "IFN-Induced PARPs—Sensors of Foreign Nucleic Acids?"

_pathogens, 2023, doi:10.3390/pathogens12030457_

Round 1

Reviewer 1 Report

The authors summarized multiple functions of IFN-dependent PARPs as a detector of virus RNA distinguishing from host RNA, and as a responder contributing anti-viral defense. The review is well designed and organized. The conclusion is supported by the detailed information. I only have some minor comments:

1.  Table 1: Readers would appreciate it if the author could add information of references either adding a column on the right side or in the Table legend.

2. To use an abbreviation, e.g., AIM2, cGAS, TRIM, please present the full term on the first use.

3. Line 311: Please define “NA binding”.

4. Line 412: In general, IFNAR1 includes IFN alpha and beta receptor 1. Please check its full term.

5.  Lines 439-441: Please use consistent font.

6. Line 469: Regarding “Parp 14”, please check official NCBI gene nomenclature.

Author Response

            We thank the reviewers for the overall positive assessment of our manuscript and appreciate their comments. Below we have addressed their concerns and comments point by point.

            We apologize for using the wrong citation style for the references throughout the text. We now integrated the MDPI citation style to our Endnote system and updated the citations and references accordingly. Further, we highlighted all changes that we made throughout the text.

Reviewer #1:

The authors summarized multiple functions of IFN-dependent PARPs as a detector of virus RNA distinguishing from host RNA, and as a responder contributing anti-viral defense. The review is well designed and organized. The conclusion is supported by the detailed information. I only have some minor comments:

            We appreciate the overall statement that our review is well designed and organized and that conclusions we draw are solid.

  1. Table 1: Readers would appreciate it if the author could add information of references either adding a column on the right side or in the Table legend.

            We thank the reviewer for this suggestion. Indeed, the addition of the relevant references will be helpful for the reader. We updated the table and added the respective information to the table.

  1. To use an abbreviation, e.g., AIM2, cGAS, TRIM, please present the full term on the first use.

            We have modified the text accordingly. Whenever an abbreviation is used for the first time, we have now added the full term with the respective abbreviation in brackets.

  1. Line 311: Please define “NA binding”.

            We thank the reviewer for attentive reading. We replaced “NA” by nucleic acid.

  1. Line 412: In general, IFNAR1 includes IFN alpha and beta receptor 1. Please check its full term.

            We adapted this accordingly in the text.

  1. Lines 439-441: Please use consistent font.

            We also adapted the font in these lines accordingly.

  1. Line 469: Regarding “Parp 14”, please check official NCBI gene nomenclature.

            Because the inhibitor studies were performed in mice, we initially wrote Parp14 to indicate the mouse protein. However, after reading this study again, we realized that the authors also refer to PARP14 and changed this in our manuscript (now line 464) accordingly.

Reviewer 2 Report

In this manuscript, Biaesch et al., summarized recent findings of IFN-induced PARPs and speculated on the underlying cause of their antiviral activities. It is well written and a significant contribution to the field. However, several points require attention and should be addressed as described below.

1.  In terms of classic nucleic acid sensors, PKR (protein kinase R) and OAS (oligoadenylate synthetase) should be included. They are missed in this work.

2. A detailed description of classic PRRs from line 70 to line 180 is not necessary. Is it helpful to IFN-induced PARPs? The reviewer suggests that the authors remove this section from the manuscript.

3. This work mainly focuses on IFN-induced PARPs. The evidence/description about that some PARPs were induced by IFN should be added. Furthermore, could the authors introduce/discuss other PARP family member?

4. The manuscript is difficult to understand due to structural problems. Please pay more attention to subheadings to ensure logic is clear.  Also, should the authors move the part of PARP13 – a sensor of viral RNA (line 181) to the back?

5. Note the font from line 439 to 443.

Author Response

            We thank the reviewers for the overall positive assessment of our manuscript and appreciate their comments. Below we have addressed their concerns and comments point by point.

            We apologize for using the wrong citation style for the references throughout the text. We now integrated the MDPI citation style to our Endnote system and updated the citations and references accordingly. Further, we highlighted all changes that we made throughout the text.

Reviewer #2:

In this manuscript, Biaesch et al., summarized recent findings of IFN-induced PARPs and speculated on the underlying cause of their antiviral activities. It is well written and a significant contribution to the field. However, several points require attention and should be addressed as described below.

            We thank the reviewer for the positive assessment of our manuscript.

  1. In terms of classic nucleic acid sensors, PKR (protein kinase R) and OAS (oligoadenylate synthetase) should be included. They are missed in this work.

            We thank the reviewer for bringing this to our attention. We have now included a paragraph describing PKR and OAS to have a more complete repertoire of sensors. We updated the table accordingly.

  1. A detailed description of classic PRRs from line 70 to line 180 is not necessary. Is it helpful to IFN-induced PARPs? The reviewer suggests that the authors remove this section from the manuscript.

            This is a justified question. However, we think it is important to keep this section in the manuscript. We only briefly outlined the binding modes of the different classical PRRs and summarized the consequences of binding to PAMPs with regard to e.g. conformational changes or activation of enzymatic activity. This seems important to us because we refer to one or the other concept at the end of the manuscript when we discuss the findings with the different PARP proteins. Also, this information is important to draw a perspective on how the IFN-regulated PARPs might interact with and be regulated by binding to foreign RNA.

  1. This work mainly focuses on IFN-induced PARPs. The evidence/description about that some PARPs were induced by IFN should be added. Furthermore, could the authors introduce/discuss other PARP family member?

            In recent years the evidence that a subset of MARylating ARTDs (PARP9 – PARP15) is responsive to IFN has been well established and is summarized in recent reviews (e.g. Fehr et al., 2020 “The impact of PARPs and ADP-ribosylation on inflammation and host-pathogen interactions”; Zhu and Zeng, 2021 “When PARPs meet antiviral innate immunity”; Luescher et al., 2022 “Intracellular mono-ADP-ribosyltransferases at the host-virus interface”). We extracted data concerning the players in ADP-ribosylation from studies performed by Shaw and colleagues for our latest review (Luescher et al., 2022). From this it becomes clear that only the expression of PARP9-PARP15 is significantly upregulated in response to IFN across different species tested (Shaw et al., 2017 “Fundamental properties of the mammalian innate immune system revealed by multispecies comparison of type I interferon responses”). We now explicitly refer to our review for detailed reading on PARPs as ISGs (line 59).

Due to this direct link to innate immunity, i.e. the domains these PARPs are equipped with and their localization in the cytoplasm, we decided to focus on these PARPs and do not discuss other PARP family members as RNA sensors.

  1. The manuscript is difficult to understand due to structural problems. Please pay more attention to subheadings to ensure logic is clear.  Also, should the authors move the part of PARP13 – a sensor of viral RNA (line 181) to the back?

            We will explain our decision of introducing PARP13 at this point. We begin with a brief introduction of the classical PRRs to provide the reader with binding preferences of these sensors and summarize the consequences of RNA binding of these PRR. PARP13 is already established as sensor of pathogenic RNA and thus classifies as a PRR (Ficarelli et al., 2021 “Targeted restriction of viral gene expression and replication by the ZAP antiviral system”). We argue that first outlining the role of PARP13 provides an excellent basis for the introduction of the other IFN-regulated PARPs, which we aimed to discuss as potential PRRs due to the domains they are equipped with and reports linking them to viral defense (which we also summarize in the manuscript). Also, data on PARP13 is rather extensive and we consider it as solid, whereas RNA binding by the other IFN-regulated PARPs is still rather preliminary. In some cases, initial evidence has been provided but clearly further work will be required to further validate these observations and to support the developed hypotheses. In the end we pick up concepts from the classical PRR and PARP13 to draw potential scenarios of the consequences of RNA binding for PARP9 – PARP12 and PARP14 and PARP15.

  1. Note the font from line 439 to 443.

            We thank the reviewer for this notion and adapted the font in these lines accordingly.

Round 2

Reviewer 2 Report

Thanks for the authors' efforts. The author has to point out that OAS2 and OAS3 function as dsRNA sensors. The reviewer has no further concerns.

Author Response

Thanks for the authors' efforts. The author has to point out that OAS2 and OAS3 function as dsRNA sensors. The reviewer has no further concerns.

            For a complete view on the OAS proteins in nucleic acid binding, indeed OAS2 and OAS3 should be included. We now added OAS2 and OAS3 to the main text, and the table accordingly and extended the references.